# Low Spike Antibody Levels and Impaired BA.4/5 Neutralization in Patients with Multiple Myeloma or Waldenstrom’s Macroglobulinemia after BNT162b2 Booster Vaccination

**DOI:** 10.3390/cancers14235816

**Published:** 2022-11-25

**Authors:** Margherita Rosati, Evangelos Terpos, Jenifer Bear, Robert Burns, Santhi Devasundaram, Ioannis Ntanasis-Stathopoulos, Maria Gavriatopoulou, Efstathios Kastritis, Meletios-Athanasios Dimopoulos, George N. Pavlakis, Barbara K. Felber

**Affiliations:** 1Human Retrovirus Section, Vaccine Branch, Center for Cancer Research, National Cancer Institute at Frederick, Frederick, MD 21702, USA; 2Department of Clinical Therapeutics, School of Medicine, National and Kapodistrian University of Athens, 117 27 Athens, Greece; 3Human Retrovirus Pathogenesis Section, Vaccine Branch, Center for Cancer Research, National Cancer Institute at Frederick, Frederick, MD 21702, USA

**Keywords:** COVID vaccine, multiple myeloma, Waldenstrom’s macroglobulinemia, neutralization, WA1, BA.1, BA.2, BA.4/5

## Abstract

**Simple Summary:**

Upon BNT162b2 mRNA vaccinations, multiple myeloma (MM) and Waldenstrom’s macroglobulinemia (WM) patient cohorts on active therapy affecting B cell development had impaired binding and neutralizing antibody (NAb) response rate and magnitude to wildtype Wuhan (WA1). Patient cohorts on-therapy had significantly lower NAb responses to SARS-CoV-2 Omicron variant BA.4/5 compared to WA1, whereas cohorts off-therapy showed a higher probability to neutralize BA.4/5 after the 3rd vaccination. The boost in NAb after the 3rd dose suggests that repeat vaccination of MM and WM patients is beneficial for several patients even under active therapy.

**Abstract:**

Patients with symptomatic monoclonal gammopathies have impaired humoral responses to COVID-19 vaccination. Their ability to recognize SARS-CoV-2 Omicron variants is of concern. We compared the response to BNT162b2 mRNA vaccinations of patients with multiple myeloma (MM, n = 60) or Waldenstrom’s macroglobulinemia (WM, n = 20) with healthy vaccine recipients (n = 37). Patient cohorts on active therapy affecting B cell development had impaired binding and neutralizing antibody (NAb) response rate and magnitude, including several patients lacking responses, even after a 3rd vaccine dose, whereas non-B cell depleting therapies had a lesser effect. In contrast, MM and WM cohorts off-therapy showed increased NAb with a broad response range. ELISA Spike-Receptor Binding Domain (RBD) Ab titers in healthy vaccine recipients and patient cohorts were good predictors of the ability to neutralize not only the original WA1 but also the most divergent Omicron variants BA.4/5. Compared to WA1, significantly lower NAb responses to BA.4/5 were found in all patient cohorts on-therapy. In contrast, the MM and WM cohorts off-therapy showed a higher probability to neutralize BA.4/5 after the 3rd vaccination. Overall, the boost in NAb after the 3rd dose suggests that repeat vaccination of MM and WM patients is beneficial even under active therapy.

## 1. Introduction

Patients with multiple myeloma (MM) or Waldenstrom’s Macroglobulinemia (WM) show low humoral and cellular responses to vaccination against SARS-CoV-2 [1,2,3,4,5,6]. Factors associated with poor response in MM include uncontrolled disease, immunosuppression, concomitant therapy, more lines of therapy, CD38 antibody-directed and B-cell maturation antigen (BCMA)-directed therapy [1,4,5,7]; WM treatment with anti-CD20-based therapies predisposes to poor response to vaccination [6,8].

The majority of studies have evaluated neutralizing antibodies (NAb) against original Wuhan-HA-1 (referred to as WA1) SARS-CoV-2 virus or against initial mutants (Alpha to Delta) after primary and booster vaccination [1,2,3,4,5,6,7,8], but there is presently few data for the efficacy of vaccination against Omicron and its subvariants in patients with hematological malignancies [9,10,11,12]. Omicron variants, including the recently emerged variants BA.4 and BA.5, which are more infectious than the original WA1 strain, are spreading throughout the globe since late 2021 [13,14,15,16,17,18,19,20]. Both primary and booster vaccinations produce low protection against Omicron variants compared to WA1 strain and initial SARS-CoV-2 variants [21,22,23]. Thus, the Omicron variants partially escape from the WA-1 induced neutralization [24,25].

The aim of this study is to evaluate magnitude and breadth of binding and neutralizing Ab induced upon vaccination in patients with MM and WM during active therapy and off therapy. The ranking of responses to several dominant virus variants was established for both patient cohorts and compared to a cohort of health care workers (HCW), all vaccinated with three doses of the BNT162b2 (Pfizer/BioNTech) mRNA vaccine, according to the same schedule.

## 2. Materials and Methods

### 2.1. Patients and Controls

This is a prospective study designed to determine the level of anti-SARS-CoV-2 Ab after three immunizations with the BNT162b2 mRNA vaccine (NCT04743388) in patients with plasma cell dyscrasias. Data from HCW who received the BNT162b2 mRNA vaccine, vaccinated during the same period, were also included in this analysis. Blood collection schedule was as follows: day of 1st (D1) and 2nd (D22) vaccination, at one month after the 2nd vaccination (D50), day of 3rd (M9) vaccination, and one month later (M10). Sera were isolated within 4 h of collection and was kept frozen at −80 °C until the day of measurement.

The study protocol was approved by the ethics committee of Alexandra General Hospital (Ref_No. 15/23_Dec_2020). The study was carried out in accordance with the Declaration of Helsinki and the International Conference on Harmonization for Good Clinical Practice standards of care. All participants provided written informed consent at study entry. Patient data were maintained according to the General Data Protection Regulation.

### 2.2. SARS-CoV-2 Antibody Measurements

In-house ELISA assays using a panel of purified full-length trimeric Wuhan (WA1) Spike and Spike-RBD proteins and a panel of Variant of Concern (VOC) Spike and Spike-RBD proteins (Delta B.1.617.2, Omicron BA.1 (B.1.1.529) and BA.2) were detailed elsewhere [12,26,27]. Ab levels were measured using eight 4-fold serial serum dilutions starting at 1:50 and endpoint titers were determined using lastX feature using GraphPad prism area-under-the curve program.

The ACE2 binding competition neutralization assay was performed using a US Food and Drug Administration–approved ELISA (cPass SARS-CoV-2 NAbs Detection Kit; Gen-Script, Piscataway, NJ, USA). A NAb titer of 30% is considered positive, whereas a NAb titer of 50% has been associated with clinically relevant viral inhibition [28,29].

Pseudotype neutralization was performed using a HIV_NL_DEnv-Nanoluc assay [30,31] carrying the 1254-AA WA1 (D616G), Delta, BA.1, BA.2 and BA.4/5 Spike proteins (AA 1-1254) as described [12,26,27]. BA.4/5 Spike, a gift from T. Hatziioannou (Rockefeller University, New York, NY, USA), also has the R683G substitution. The highest serum concentration analyzed was a 1:40 dilution and 8 four-fold serial dilutions were tested. Two days later, the luciferase levels were measured in the cell extracts as ID50 (50% Inhibitory Dose) calculated using GraphPad Prism version 9.2 for MacOS X (GraphPad Software, Inc., La Jolla, CA, USA). The NAb ID50 threshold of quantification in this assay is 0.5 log and the threshold of detection is 0.1 log.

### 2.3. Statistical Analysis

There was no randomization performed due to the nature of the study. Statistical analysis was performed with the GraphPad Prism 9.2.0 Software for mac using GraphPad Software, San Diego, CA, USA. For comparisons of between HCW and the different cohorts, the *p* values are from non-parametric ANOVA (Friedman test). Non-parametric *t* test (Wilcoxon) was applied to compare two groups. Simple linear regression was determined by GraphPad Prism. A value *p* < 0.05 was considered statistically significant.

## 3. Results

### 3.1. Patients’ Characteristics

This is a prospective study to monitor the kinetics and breadth of anti-COVID-19 humoral immune responses upon SARS-CoV-2 BNT162b2 (Pfizer/BioNtech) mRNA vaccination in cohorts of MM and WM patients (NCT04743388). The vaccine was administered 3 times per protocol on Day 1 (D1), day 22 (D22) and month 9 (M9). Sera were collected before vaccine administration on the days of vaccination (D1, D22, M9) and one month after the 2nd (D50) and 3rd (M10) vaccination. The patient samples were distinguished in five cohorts (WM, group (G) G1 and G2; MM G3 to G5) based on the active or prior therapy (Table 1). MM patients on active therapy were further distinguished in two groups (G3 and G4) based on the types of treatment (Table 1): WM patients (n = 20) comprise G1, with 11 patients on active therapy, including BTK inhibitor (BTKi)-based or anti-CD20 rituximab-based regimens) and G2, with 9 patients off-therapy. MM patients (n = 60) comprise G3a, with 17 patients treated with anti-CD38; G3b, with 3 patients treated with anti-BCMA; G4, with 20 patients on other therapies [proteasome inhibitors (PI) or immunomodulatory drugs (IMiD) without anti-CD38 or anti-BMCA], and G5, with 20 MM patients off-therapy. WM and MM cohorts off-therapy (G2 and G5, respectively) were enrolled at a median time of 30 months (range 10–86) and 37 months (range 1–126), respectively, after prior treatment. Patients who received treatment with combination of an anti-CD38 with PI/IMiD were categorized in the “anti-CD38 subgroup”, because the anti-CD38 treatment was considered to have a particular impact on humoral responses post vaccination. The patient cohorts were compared to a cohort of BNT162b2 mRNA vaccinated HCW (n = 37) as controls.

The patient demographics showed a similar age distribution and a disparate gender distribution with more female patients in the WM cohort and more male patients in the MM cohort (Table 1). The patient cohorts also differed in age from HCW. One patient in WM off therapy cohort (G2) tested positive for anti-Spike Ab and NAb at day 1. Two patients in the HCW cohort were COVID-19 positive at study entry. Because this study focused mainly on antibody responses obtained after the 3rd dose, there was no reason for exclusion of these individuals, because we have previously shown that prior exposure to COVID-19 results in a strong boost after the 1st dose of a vaccine; thereafter, antibody levels were not different from COVID-19 negative vaccine recipients [32].

### 3.2. Kinetics of Development of Neutralizing Anti-WA1 Spike Antibodies

All samples were evaluated using the semi-quantitative cPass SARS-CoV-2 NAb detection assay for sera collected overtime monitoring the effect of the 1st, 2nd, and 3rd doses (Figure 1A–G). The ability to inhibit interaction of Spike-RBD with ACE-2 greatly increased upon the 2nd and the 3rd dose, pointing to the importance of the booster vaccination. The HCW cohort (Figure 1A), the cohorts of WM off-therapy (Figure 1C), MM on PI-/IMiD-based therapies (Figure 1F) and MM off-therapy (Figure 1G) showed high response rates >90% at M10, considering values above the clinically relevant inhibition threshold. The MM cohort treated with anti-CD38 (G3a, 9 of 17 responders, Figure 1D) showed greatly impacted ACE2 binding inhibition (~50% response rate) while the cohorts of WM on-therapy (G1, 1 of 10 responders, Figure 1B) and MM treated with anti-BCMA (G3b, 1 of 3 responders, Figure 1E) showed the lowest responses (~10–30% response rate). Therefore, although all cohorts benefited from the booster vaccination, different treatment regimens including BTKi or rituximab (WM, G2), and anti-CD38 or anti-BCMA regimens (MM, G3) significantly impaired the development of vaccine-induced neutralizing antibodies (NAb) and consequently negatively impact on the patients’ ability to neutralize SARS-CoV-2.

To better characterize and quantitate the vaccine induced antibody responses in the different patient cohorts, the magnitude and breadth of anti-Spike binding antibodies (Ab, Figure 2 to Figure 3) and neutralizing antibody (NAb, Figure 4 and Figure 5) responses were measured.

### 3.3. Distinct Anti-Spike Antibody Responses to the BNT162b2 mRNA Vaccine in WM and MM Patients Compared to HCW

The development of anti-Spike Ab was monitored in serum samples collected on D50 (one month after administration of the 2nd dose), on the day of the 3rd vaccination (M9) and one month later (M10) using an in-house ELISA, measuring binding to the trimeric wildtype Wuhan (WA1) Spike protein. The responses of the WM cohort (G1 Figure 2B and G2; Figure 2C) and the MM cohorts (G3 to G5, Figure 2D–G) were compared to a cohort of HCW (Figure 2A). The patients were either on active therapy (WM: G1, Figure 2B; MM: G3a and G3b, Figure 2D,E and G4, Figure 2F) or off-therapy (WM, G2, Figure 2C; MM, G5, Figure 2G). Importantly, we noted a remarkable broad range in anti-Spike Ab response magnitude in the different patient cohorts (Figure 2B–G) compared to HCW (Figure 2A), at all three time points analyzed.

A 100% response rate was found both at one month after the 2nd (D50) and one month after the 3rd (M10) dose in HCW (Figure 2A), the cohorts of WM off-therapy (Figure 2C), MM on PI/IMiD-based therapy regimens (Figure 2F), and MM off-therapy (Figure 2G). In contrast, a strongly impaired response rate was found even after the 3rd dose in WM on-therapy (Figure 2B; 5 of 10 patients), the MM cohorts on anti-CD38 (Figure 2D, 15 of 17 patients) or anti-BCMA (Figure 2E, 2 of 3 patients) regimen. Although the response rate improved in the anti-CD38 treated MM cohort (Figure 2D, from 12 to 15 of 17 patients), the booster vaccination did not improve the response rate in the WM on-therapy (Figure 2B) and MM cohorts on anti-BCMA therapy (Figure 2E). The MM cohort on PI/IMiD-based therapy (G4, Figure 2F) and the MM off-therapy (G5, Figure 1G) showed response rates like HCW.

The BNT162b2 booster vaccination (3rd dose) resulted in a median 16-fold increase in Ab titer, except in the WM cohort on active therapy where only few patients (3 of 10) showed an increase (Figure 2B). In comparison to the responses upon the 2nd vaccination (D50), the Ab titers were significantly higher (Wilcoxon *t* test) after the 3rd dose (M10) for HCW (Figure 2A), the different MM cohorts on active therapies (Figure 2D,F) as well as MM off-therapy (Figure 2G). In contrast, the overall Ab levels in the WM cohorts on-therapy (Figure 2B) or off-therapy (Figure 2C), and anti-BCMA-treated MM cohort (Figure 2E) did not significantly increase, despite few individuals showed anamnestic responses. This may reflect power limitation due to the cohort sizes.

Comparison of the Spike antibody titers at month 10 (Figure 2H) showed the wide range of responses in the patient cohorts clearly distinct from the HCW. The responses were lower in the different cohorts but did not reach significance (ANOVA) in the off-therapy WM (G2) and MM (G5) cohorts. Of note, several patients in cohorts under active treatment (WM G1, MM G3) failed to mount Ab response and the patients with positive responses had greatly reduced Ab titers.

Together, a benefit of the booster vaccination was found in the increased anti-Spike Ab levels in all cohorts, although the extent varied. In contrast to the more homogeneous responses in HCW, a great inter-cohort difference was noted even in patient cohorts off-therapies, leading to a more heterogeneous response. Treatments affecting B cell development [BTKi or anti-CD20 (Figure 2B), anti-CD38 (Figure 2D), anti-BCMA, (Figure 2E)] severely interfered with Ab development. However, anamnestic responses seen in several patients after the 3rd vaccination greatly supports the need for continued vaccinations in these patients, even while under active therapy.

### 3.4. Similar Anti-Spike Antibody Response Breadth to the BNT162b2 mRNA Vaccine in WM and MM Patients Compared to HCW

The different cohorts were further evaluated for their ability to recognize a panel of trimeric Spike proteins including Delta, Omicron BA.1 and BA.2 (Figure 3A) with amino acid (AA) changes found within the RBD as well as in the N-and C terminal domains (NTD, CTD) and the Heptad Region 1 (HR1).

Serum samples showing positive WA1 Ab responses (see Figure 2) were analyzed after the 2nd and after the 3rd vaccination (Figure 3B–G, left and right panels, respectively). Compared to WA1 responses, the HCW cohort (Figure 3B) showed significant lower recognition of Delta and the Omicron variants BA.1 and BA.2, after the 2nd and the 3rd dose. Similarly, the WM and MM patient cohorts showed a similar response breadth with stronger recognition of Delta but significantly lower binding to BA.1 and BA.2, respectively. Of note, the samples were collected in October 2021 and the Omicron variant was first reported in Greece in December 2021, thus, the measurement of anti-Omicron Spike responses reflected cross-reactivity from the Wuhan Spike vaccine.

Together, although several of the WM and MM cohorts had lower vaccine-induced humoral immune responses (Figure 2), they showed a similar breadth and ranking of Ab responses, comparable to those of HCW cohort. These data showed that the magnitude, but not the cross-reactivity of the Spike Ab was affected by certain cancer (WM, MM) and treatment regimens.

### 3.5. Distinct Ab Magnitude to Spike and Spike-RBD Proteins of WA1, Delta, BA.1 and BA.2

Spike and Spike-RBD Ab levels were compared after the 3rd dose (M10) in HCW and the different patient cohorts (Appendix A). Comparison of antibody titers showed a median 2-fold (range 1–4.6) lower binding to Spike-RBD compared to the trimeric Spike proteins testing WA1 and the different proteins in the panel (Delta, BA.1 and BA.2) (Appendix A). Thus, the Ab induced in HCW and in the different patient cohorts have similar ability to recognize the panel of Spike-RBD proteins. However, the cohorts of WM on active therapy (G1) and of MM on anti-CD38 and anti-BCMA regimens (G3a and G3b) were severely affected in both vaccine-induced response rates and/or magnitude resulting in their severely reduced ability to bind to Spike-RBD. Since WA1 Spike-RBD endpoint titers of <3000 are predicted to lack neutralization ability in the pseudotype virus assay [12,27], 50–90% of patients (cohorts of WM on-therapy; MM on anti-CD38 or anti-BCMA regimens) were predicted to also have poor neutralization capabilities.

All patient cohorts developed lower magnitude of Spike Ab responses with a similar but lower proportion of Spike-RBD responses, comparable to HCW. Cohorts including WM off-therapy (G2), MM off-therapy (G5) as well as MM on PI or IMiDs regimens (G4) showed a response magnitude and breadth comparable to HCW, although their responses range was broader. Of concern were WM on-therapy (G1, BTKi or rituximab) and MM (G3, anti-CD38, anti-BCMA) cohorts with a substantial number of patients (50–90%) who developed very low Spike-RBD antibodies.

### 3.6. Kinetics of Development of Neutralizing Anti-WA1 Spike Ab

Because the cPASS NAb assay (Figure 1) does not allow quantitative assessment of the positive responses, we further evaluated the neutralization ability of the vaccine induced Ab by employing a functional neutralizing assay. Patient sera were tested for their ability to inhibit infection of ACE-expressing HEK293 cells by HIV-derived lentiviruses carrying a panel of Spike proteins. The neutralization capability of WA1 D614G Spike carrying pseudotype virus was measured as Infectious Dose 50 (ID50) (Figure 4). Responses at threshold (serum dilution 1:40; ID50 0.5 log) were considered negative. Applying a selection threshold, samples with Spike-RBD Ab titer >3000 were included (Appendix A), because they were likely to show measurable neutralization in our functional assay [12,27]. In further support of this notion, we tested seven sera with Spike-RBD Ab levels below the selection cut-off (red symbols below the threshold line) which were indeed negative, along our prediction (Figure 4, red square symbol).

Comparison of the two assays [cPass SARS-CoV-2 NAb detection assay (see Figure 1) and pseudotype NAb assay (Figure 4)] were in good agreement in scoring samples with positive responses (Appendix A). The pseudotype neutralization assay further showed a broad range of NAb responses in the different WM and MM cohorts compared to HCW (Figure 4). Compared to HCW, the NAb levels were significantly (ANOVA) lower in all cohorts, except WM off-therapy (G2) and showed a trend for MM off-therapy cohort (G5), with both showing a ~80–90% response rate. These data overall reflected their binding Ab responses reported in Figure 2H. As predicted, patients with low Spike-RBD Ab titers scored negative in the NAb assay. The WM cohort on-therapy (G1) showed the lowest NAb responses with only one patient (treated with the BTKi Zanabrutinib) of 11 patients developing NAb. The MM cohorts on anti-CD38 or anti-BCMA regimens (G3) also showed a disparate outcome with a low response rate of ~50% with stronger NAb development in half of the patients and lack thereof in the other half.

Individuals with low Spike-RBD Ab responses below our predictive threshold did not mount neutralizing Ab (evaluated in 2 different assays). As a result, they lack the ability to be protected COVID-19 infection despite being vaccinated and boosted. This is especially critical for the cohorts (G1 and G3: WM, on BTKi-based or rituximab-based regimens; MM on anti-CD38 or anti-BCMA therapies) where most individuals (90% and 45%, respectively) have low or no NAb responses. In the WM cohort off-therapy (G2) as well and MM cohorts on PI/IMiD-based regimens (G4) or off-therapy (G5), few individuals (10–20%) failed to induce WA1 NAb. In addition, the failure of developing strong vaccine-induced immune responses at peak (one month post 3rd dose) is of further concern since the responses are expected to contract.

### 3.7. Neutralization Breadth with Impaired Responses to BA.4/5

The sera with positive WA1 NAb responses (Figure 4) were further tested for their ability to recognize Delta, Omicron variants BA.1, BA.2 and BA.4/5 (BA.4 and BA.5 share the same Spike sequence) in the pseudotype neutralization assay (Figure 5). The HCW cohort showed that the ability to neutralize Delta and Omicron variants was significantly reduced (Figure 5A). The WA1 vaccine-induced Ab showed ranking of neutralization of Omicron BA.1 > BA.2 >> BA.4/5, rendering the BA.4/5 carrying pseudotype virus significantly most difficult to neutralize. At one month after the 3rd dose, 6 of 37 (16%) HCW failed to neutralize BA.4/5, despite robust neutralization of Delta and BA.1 and BA.2.

Overall similar conclusions were drawn from the analysis of different WM and MM cohorts (Figure 5B–E). As noted for WA1 Spike neutralization, a broad range of NAb responses to the panel of pseudotyped viruses was found within different patient cohorts. Several patients with low NAb to WA1 (ID50 < 2.5 log) showed strongly reduced ability to neutralize the Delta Spike carrying pseudotyped virus and very poor or no ability to neutralize Omicron variants BA.1, BA.2 and especially BA.4/5. In general, sera lacking ability to neutralize BA.1 and/or BA.2 also failed to neutralize BA.4/5 (Figure 5B–E). A drastically reduced number of patients shows BA.4/5 neutralization. No BA.4/5 NAb was found in the single WM patient on-therapy (Figure 5B, open triangle symbol) and in the single MM patient on anti-BCMA therapy (Figure 5C, open triangle symbol), although both individuals showed NAb titers to WA1, albeit at lower levels.

To understand the full breadth of NAb responses covering WA1 and BA.4/5, as the most diverged VOC and currently the most spreading virus, we evaluated these responses in the different cohorts (Figure 5F). Analysis of HCW showed a high response rate of 84% with WA1 and BA.4/5 NAb. In contrast, in the combined WM cohorts on- and off-therapy (G1 and G2) this applied to only 22% (4 of 18 patients), with none of 10 patients on-therapy and 4 of 8 patients (6 off-therapy showing responses. In the MM cohort on-therapy, 15% (G3) and 25% (G4) could neutralize both Spike proteins. On the other hand, the MM cohort off-therapy (G5) showed a higher response rate of 42% of patients with WA1 and BA.4/5 NAb. Together, these data show that the patient cohorts are vulnerable with many patients not protected against Omicron.

To evaluate a possible connection between magnitude of binding Ab and their neutralization capabilities, we performed correlation analyses between anti-WA1 Spike-RBD antibody titers and the cross-reactive neutralization abilities measured at month 10 (Figure 5G). We further performed such correlations also using BA.2 Spike-RBD Ab responses as surrogate, due to the closest relation of BA.2 and BA.4/5 Spike (differing by 6 AA) (Figure 5H). Excellent correlations were found for the HCW cohort (Figure 5G,H, top panels), supporting the notion that high levels Spike-RBD Ab at one month post 3rd dose (M10) predicted strong cross-neutralization. However, the failure of 16% of the sera to neutralize BA.4/5, even at the peak (M10), raised a concern. Similar strong correlations to WA1, Delta, and the Omicron variants BA.1, BA.2 and BA.4/5 were found in the WM (Figure 5G, middle panel) and MM (Figure 5H, bottom panel) cohorts, irrespective of the therapy status. The slopes of the linear regression curves comparing WA1 NAb and BA.4/5 NAb were significantly different for all the cohorts supporting the distinct and poorer neutralization of BA.4/5, visualized by the steeper slopes. Of note, sera with WA1 Spike-RBD binding Ab titers <5 log had >90% probability of failure to neutralize BA.4/5.

Together, these data show that strong ELISA WA1 Spike-RBD Ab titers (>5 log) in HCW and the WM and MM cohorts were good predictors for the ability to potently neutralize not only WA1 but also the most diverged BA.4/5. For this reason, the WM and MM cohorts having lower binding Ab titers were of great concern, and indeed, they showed overall poor neutralization of the currently circulating SARS-CoV-2 with BA.4 and BA.5 Spike. Considering that these responses were measured at peak (M10) and they will naturally contract, these findings predict that even patients who developed low BA.4/5 NAb are at high probability to lose this response and are at high risk of SARS-CoV-2 infection.

## 4. Discussion

Knowledge of immune response and neutralization capacity against Omicron variants after vaccination in patients with MM and WM is of great importance for the prevention of severe COVID-19. We compared the Spike and Spike-RBD Ab magnitude and breadth and its ability to cross-neutralize WA1, Delta, and the Omicron variants BA.1, BA.2 and BA.4/5 following BNT162b2 in MM/WM patients and in HCW as reference. Analysis of HCW showed potent neutralization of WA1 and significantly lower NAb to Delta, BA1, BA.2 and BA.4/5, similar to previously reported data [25,33,34,35,36].

We showed that WM and MM cohorts, independent of being on active therapy or off-therapy, showed a broad response range. In general, cohorts on active therapy including regimens interfering with antibody development (BTKi, anti-CD20, anti-CD38, anti-BCMA) showed lower response rate and lower magnitude. WM and MM cohorts having prior therapy showed improved responsiveness to the BNT162b2, although the response range was broad. These results expand and confirm previous reports on the adverse role of B-cell directed therapies on NAb activity against WA1 following booster COVID-19 vaccination [8,36]. Another recent study also showed similar results in patients with hematological cancer who received a booster mRNA-1273 shot at 5 months after the primary mRNA-1273 vaccination [37].

Τhe low levels of binding antibody found in several of the vaccine recipients raises a great concern. We previously showed direct correlation of WA1 Spike- and Spike-RBD titers and WA1 NAb titers in cohorts of BNT162b2 vaccinated or vaccinated transplant patients with hematological malignancies or convalescent individuals and well as in WA1 DNA and DNA + Protein vaccinated macaques [12,26,32,38,39]. We report that several individuals in the WM and MM cohorts with low Spike-RBD Ab responses fail to mount NAb to WA1 and especially to BA.4/5. Thus, low WA1 Spike-RBD responses were predictive of low/lack of neutralization. Because we and others have also reported excellent correlation between the pseudotype NAb assay and live-virus neutralization [17,26,38], these data predict also lack of live-virus neutralization of the BA.4 and BA.5. Interestingly, except in the WM cohort on-therapy, there were patients who mounted high anti-Spike Ab responses, with similar quality like those induced by HCW, and these patients were able to neutralize the BA.4/5 pseudotyped virus.

## 5. Conclusions

The picture that emerged from this analysis shows a more complex vaccine effectiveness in MM/WM patient cohorts as compared with HCW. Of note, even if the responses were low after the 2nd dose, the majority of, individuals showed anamnestic responses after the 3rd dose, which is in line with the current literature on immunocompromised vaccine recipients [9,37,40,41]. T-cell responses may present distinct dynamics after each vaccine shot and may contribute to the benefit derived from anamnestic vaccination [42,43,44,45,46], however, evaluating the cellular response to BNT162b2 was beyond the scope of our study. Considering, also the lack of persistence of the Spike antibody responses and NAb responses [8,36,47,48,49,50,51], it remains important to enroll all patients in future booster vaccination regardless of their humoral response after the 3rd dose. Patients with MM or WM, especially those on active treatment, should be highly considered for pre- and post-exposure prophylaxis with monoclonal antibodies against SARS-CoV-2 in the era of Omicron variant predominance [52,53].

## Figures and Tables

**Figure 1 cancers-14-05816-f001:**
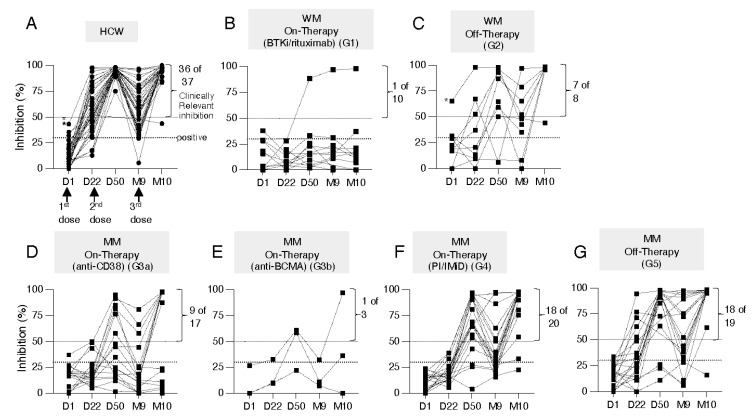
NAb development and kinetics among HCW and patient cohorts. (**A**–**G**) Kinetics of NAb development using the commercial ELISA (cPass SARS-CoV-2 NAb Detection Kit; Gen-Script, Piscataway, NJ, USA) measuring the % ACE2-binding inhibition of WA1 Spike. Inhibition >30% is considered positive, inhibition >50% has been associated with clinically relevant viral inhibition [28,29], indicated by dotted lines. (**A**) HCW and (**B**,**C**) WM (G1, G2) and (**D**–**G**) MM (G3a, G3b, G4, G5) patient cohorts were analyzed at D1, D22 (day of 2nd dose), D50, M9 (day of 3rd vaccination) and one month later at M10. The patient cohorts on active therapy ((**B**), G1; (**D**), G3a; (**E**), G3b; (**F**), G4) and off-therapy ((**C**), G2; (**G**), G5) are indicated. Asterisks denote one WM patient off-therapy and two HCW individuals with SARS-CoV-2 infection before study entry. The number of samples scoring positive within clinically relevant range of inhibition is given.

**Figure 2 cancers-14-05816-f002:**
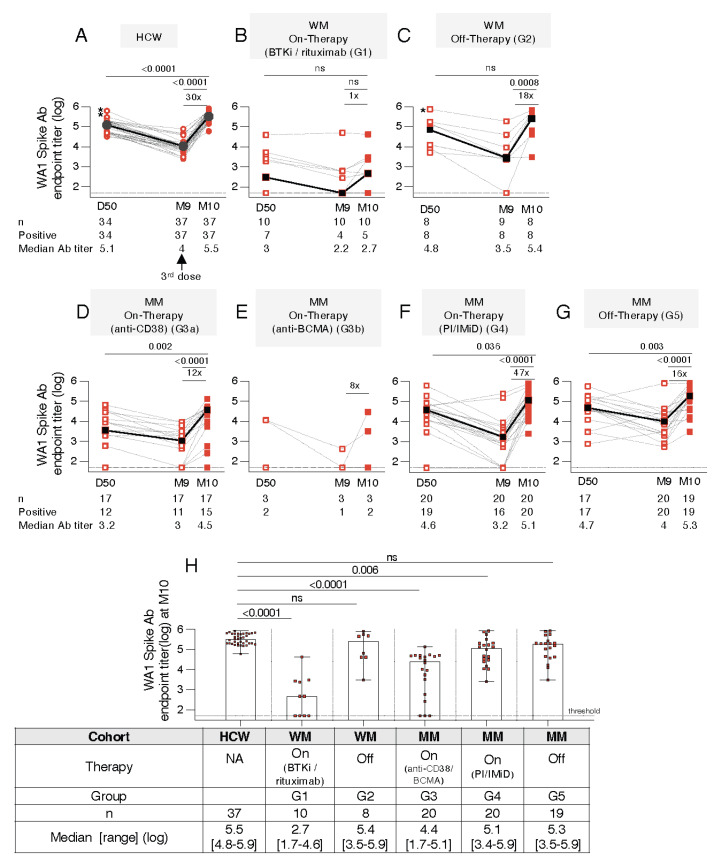
Anti-WA1-Spike antibody levels after the 2nd and 3rd BNT162b2 mRNA. vaccination. Anti-Spike antibody endpoint titers (log) were measured against purified trimeric Wuhan strain (WA1) Spike by an in-house ELISA assay. The measurements were performed overtime at one month after 2nd dose (D50), the day of the 3rd dose (M9) and one month later (M10). (**A**–**G**) Data are shown for the (**A**) vaccinated HCW; (**B**) WM (G1) and (**D**–**F**) MM cohorts (G3a, G3b, and G4) enrolled while on active therapy and (**C**) WM (G2) and (**G**) MM (G5) cohorts off-therapy, as described in Table 1. The number of analyzed samples, the positive responders and median Ab titers are given. For G1, 10 samples were available; for G2, D50 or M10, respectively, were not available for two patients. A titer at or below the threshold of the assay is entered as 50. Asterisks denote one WM patient off-therapy (panel C) and two HCW individuals (panel A) with SARS-CoV-2 infection before study entry. Median values are given in black symbols and black lines. The *p* values are from *t* test (Wilcoxon). (**H**) Comparison of Spike antibody endpoint titers at M10. The data for G3a and G3b were combined and reported as G3. The *p* values are from ANOVA (Friedman test) for the comparisons to the HCW Ab responses. Median antibody titers and range are given. ns—no significant difference.

**Figure 3 cancers-14-05816-f003:**
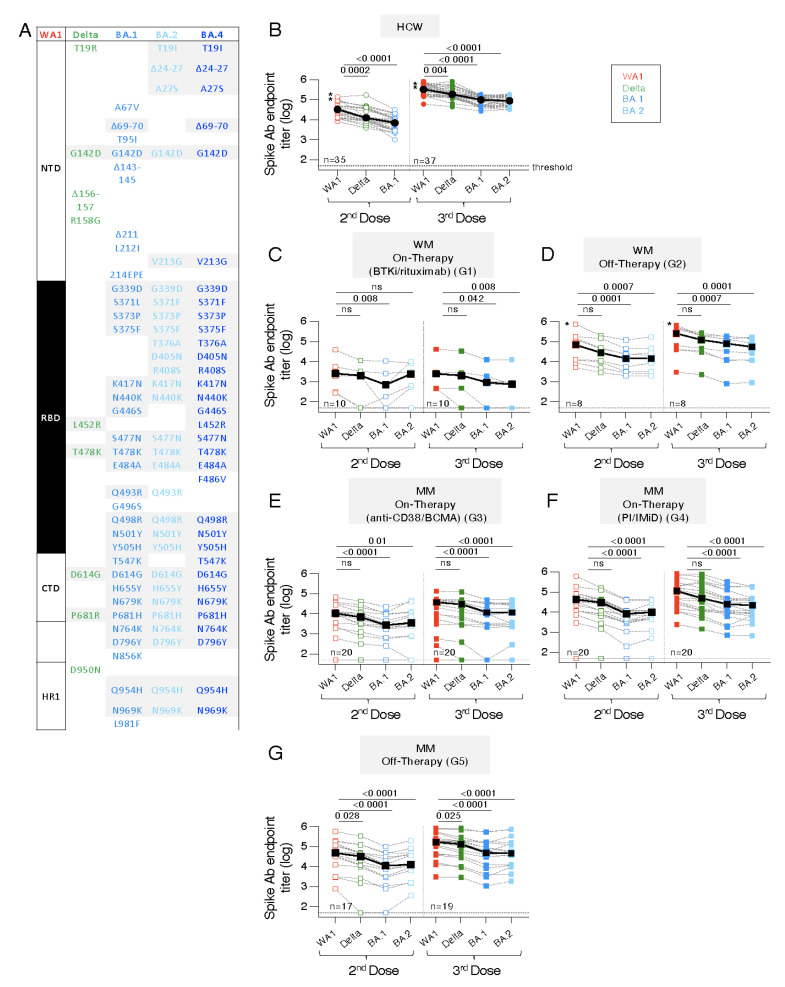
Breadth of anti-Spike antibody responses in patient cohorts and control BNT162b2 mRNA vaccine recipients. (**A**) Cartoon depicting AA changes comparing the original WA1 Spike proteins to a panel of Variants of Concern (VOC) including Delta (9 positions), Omicron variants BA.1 (34 positions), BA.2 (29 positions) and BA.4/5 (33 positions). Grey shading denotes shared changes. The following Spike regions are denoted: Spike S1 comprising NTD, N-terminal domain; RBD, Receptor Binding Domain; and CTD, C-terminal domain; and the Spike S2 with the Heptad Region 1 (HR1). (**B**–**G**) Comparisons of anti-Spike antibody responses to WA1 and Delta, Omicron BA.1 and BA.2, performed at one month after the 2nd (D50) and 3rd dose (M10) in (**B**) HCW and the different (**C**,**D**) WM and (**E**–**G**) MM patient cohorts. Median values are given in black symbols and black lines. A titer at the threshold of detection is entered as 50 (log 1.6). Asterisks denotes WM patient and HCW with SARS-CoV-2 infection before study entry. The *p* values are from ANOVA (Friedman test) for the comparisons to the WA1 NAb responses. ns—no significant difference.

**Figure 4 cancers-14-05816-f004:**
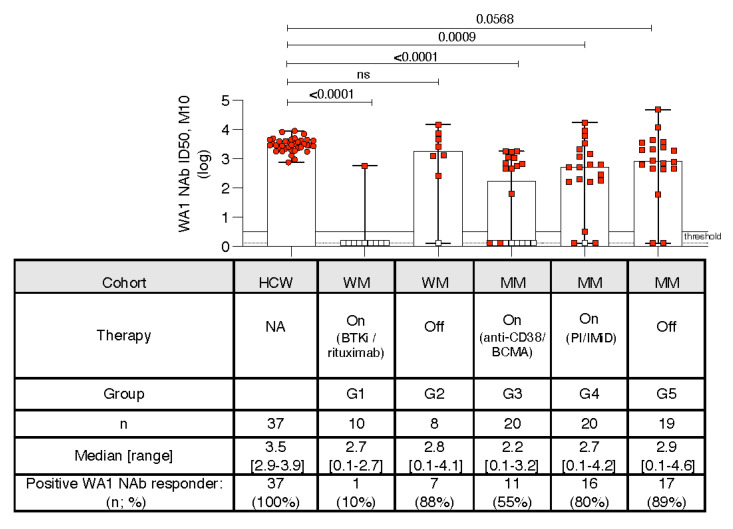
Breadth of NAb in HCW and patient cohorts. Neutralization was performed with a functional test using a pseudotype HIV_NL_ ΔEnv-Nanoluc assay [30,31] carrying WA1 (D614G) Spike and the 50% Inhibitory Dose (ID50) was determined as described [12,26,27]. Data are shown for the HCW and the WM and MM patient cohorts after the 3rd dose at month 10. Box plot and whiskers (range) are shown. The threshold of quantification (0.5 log; solid line; considered negative) and the threshold of detection (0.1 log, dotted line) are indicated. Open square symbols denote samples with Spike-RBD titers <3000, which were not tested but predicted to be negative and were entered as 0.1. The number of samples tested, median and range of responses, and the number and % of positive responders are given. The *p* values are from ANOVA (Friedman test) for the comparisons to the HCW NAb responses. ns—no significant difference.

**Figure 5 cancers-14-05816-f005:**
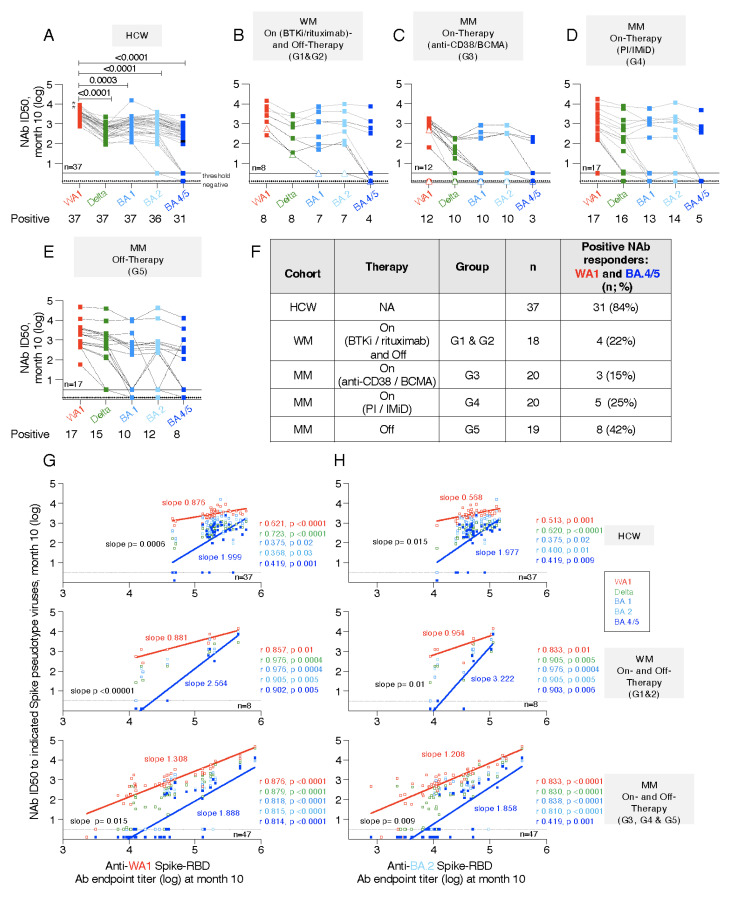
Antibody neutralization breadth in HCW and different patient cohorts after the 3rd dose. Sera with Spike-RBD endpoint titers >3000 (Appendix A) were tested for their ability to neutralize pseudotyped HIV_NL_ΔEnv-Nanoluc carrying Delta, Omicron BA.1, BA2 and BA.4/5 Spike proteins using samples collected at month 10. (**A**) HCW cohort with *p* values from ANOVA (Friedman test) for the comparisons to the WA1 NAb responses. Asterisk denotes the two persons with SARS-CoV-2 infection prior to vaccination. (**B**) WM cohort (G1 & G2) [on- (triangle) and off-(square) therapy] and (**C**,**D**) MM cohort on (**C**) anti-CD38 or anti-BCMA (G3) and (**D**) on PI-/IMiD-based therapies (G4) and (**E**) MM off-therapy (G5). (**F**) Comparison of numbers of sera in the different groups positive for WA1 and BA.4/5 NAb at month 10. Data are from panels A–E. (**G**,**H**) Correlations of (**G**) WA1 and (**H**) BA.2 Spike-RBD endpoint Ab titers and NAb ID50 titers to WA1, Delta, BA.1, BA.2 and BA.4/5 measured at month 10. Spearman r and *p* values and samples numbers are listed. The slopes of the linear regressions from the correlations (WA1 [red lettering], BA.4/5 [blue lettering]) are given. Difference (*p* values) between slopes of the linear regressions comparing WA1 and BA4.5 and the NAb measurements are given.

**Table 1 cancers-14-05816-t001:** Characteristics of Vaccine Recipients.

Cancer	Patient Group (G)	Enrolled (n)	Active Therapy	Median Time Off-Therapy, Month [Range]	Gender (n)		Age
					Male	Female	Median [Range]
WM	G1	11	ON ^1^		4	7	70
							[43–86]
WM ^2^	G2	9	OFF ^3^	30	3	6	82
				[10–86]			[62–88]
MM under anti-CD38-based regimen	G3a	17	ON		8	9	69
							[44–81]
MM under belantamab mafodotin (anti-BCMA) only regimen	G3b	3	ON		2	1	67
							[65–73]
MM under PI and/or IMiD treatments ^4^	G4	20	ON		12	8	67
							[48–86]
MM ^5^	G5	20	OFF ^6^	37	13	7	69.5
				[1–126]			[44–91]
Controls							
HCW ^7^	N/A ^8^	37	N/A ^8^		13	24	54
							[27–69]

^1^ Treatment at the time of vaccination included: Bruton-kinase inhibitor (BTKi)-based regimens without rituximab (n = 6); rituximab-based therapies (n = 5). ^2^ Includes one patient positive for COVID-19 Spike and Spike-RBD Ab and low level NAb at study entry. ^3^ Prior therapies include: rituximab-based regimens (n = 7), chemotherapy only (n = 1) and bortezomib-based therapy (VCD, n = 1). ^4^ Anti-myeloma therapies at the time of vaccination included: proteasome inhibitor (PI)-based regimens (n = 6), Immunomodulatory drug (IMiD)-based regimens (n = 1), both PI plus IMiD-based regimens (n = 13). ^5^ Includes one patient with plasmatocytoma, no prior therapy. ^6^ Prior anti-myeloma therapies included: proteasome inhibitor (PI)-based regimens (n = 1), Immunomodulatory drug (IMiD)-based regimens (n = 3), both PI plus IMiD-based regimens (n = 16). ^7^ Cohort of health care workers (HCW), including two individuals positive for COVID-19 by PCR, Spike Ab and NAb at study entry. ^8^ N/A, not applicable.

## Data Availability

The data that supports the findings of this study are available in the manuscript and Appendix A of this article.

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
