# Peer review of "Low Spike Antibody Levels and Impaired BA.4/5 Neutralization in Patients with Multiple Myeloma or Waldenstrom’s Macroglobulinemia after BNT162b2 Booster Vaccination"

_cancers, 2022, doi:10.3390/cancers14235816_

Round 1
Reviewer 1 Report
Rosati et. al. described Low Spike Antibody Levels and Impaired BA.4/5 Neutraliza in Patients with MM or WM After BNT162b2 Booster Vaccination. As describe by authors, there are some similar reports. Therefore, the originality of this work is low.
Major point:
For the better understanding of readers, authors should expain about antibodies aginst spike proteins and RBD regarding Figure 3.
Lines 312-315: Please show data regarding "Comparison of the two assays [cPass SARS-CoV-2 NAb detection assay (see Figure 1) and pseudotype NAb assay (Figure 4)] were in good agreement in scoring samples with positive responses."
Minor points:
Abstract: RBC means "receptor-binding domain"? Please spell it out.
In figure2, I cannot recognize two HCW individulas marked with asterisks.
In figure3, VOC, NTD, RBD, CTD and HR1 should be spelled out. Readers have to understand figures by themselves.
In figure 4, what red color means?
Author Response
We thank the reviewer for their comments and have provided the point-by-point response below:
Major point:
- For the better understanding of readers, authors should explains about antibodies aginst spike proteins and RBD regarding Figure 3.
OUR RESPONSE: We replaced Figure 3 in the revised manuscript, and we added a legend of the different Spike proteins used and we also added labelling to the X-axis of panel C, D, E.
- Lines 312-315: Please show data regarding "Comparison of the two assays [cPass SARS-CoV-2 NAb detection assay (see Figure 1) and pseudotype NAb assay (Figure 4)] were in good agreement in scoring samples with positive responses."
OUR RESPONSE: We added a new Figure S2 (lane 329) showing comparison of the two assays in the revised manuscript. This comparison shows an overall agreement among the assays and further shows that the pseudotype NAb assay has the power to better distinguish the neutralization capability.
Minor points:
Abstract: RBC means "receptor-binding domain"? Please spell it out.
OUR RESPONSE: We corrected this in the revised manuscript (lane 35)
In figure2, I cannot recognize two HCW individulas marked with asterisks.
OUR RESPONSE: We provide a revised Figure 2 and used a bigger font for the asterisks in panels A and C. We clarified this in the legend to Figure 2 in revised manuscript (lane 206):
Asterisks denote one WM patient off-therapy (panel C) and two HCW individuals (panel A) with SARS-CoV-2 infection before study entry.
In figure3, VOC, NTD, RBD, CTD and HR1 should be spelled out. Readers have to understand figures by themselves.
OUR RESPONSE: We clarified this in the legend to Figure 3 in revised manuscript.
Lane 282: Variants of Concern (VOC)
Lane 284-286: The following Spike regions are denoted: Spike S1 comprising NTD, N-terminal domain; RBD, Receptor Binding Domain; and CTD, C-terminal domain; and the Spike S2 with the Heptad Region 1 (HR1).
In figure 4, what red color means?
OUR RESPONSE: Throughout the manuscript the red color is used to show the antibody data to the original Wuhan Spike.
Reviewer 2 Report
This paper investigates the antibody titre of COVID19 over time after vaccination in MM and WM patients, except for the Omicron variant, which is not very new, as it has been reported by several groups including the authors group. Although the number of patients is small, the results obtained are generally valid and convincing.
The authors categorise MM patients into those who received CD38 antibody therapy and those who received PIs or IMids, but as the combination of CD38 antibody therapy and PIs or IMids is common in recent MM treatment, it would be desirable to provide details on which group the patients who received these treatments together were categorised.
Figure 5 was difficult to interpret and it was not clear what the number on the horizontal axis represented. The interpretation of the slope of the linear regression was also difficult to interpret.
Furthermore, this report shows that most post-vaccination antibody titres, including HCW, have low neutralising activity against Omicrons. Considering that many infections with Omicorn variants have occurred in healthy individuals, who are also thought to have high antibody titres after booster, It is desirable to show the clinically relevant level of neutralising titres protective against Omicron variants.
Author Response
We thank the reviewer for their comments and have provided a point-by-point response below:
This paper investigates the antibody titre of COVID19 overtime after vaccination in MM and WM patients, except for the Omicron variant, which is not very new, as it has been reported by several groups including the authors group. Although the number of patients is small, the results obtained are generally valid and convincing.
- The authors categorise MM patients into those who received CD38 antibody therapy and those who received PIs or IMids, but as the combination of CD38 antibody therapy and PIs or IMids is common in recent MM treatment, it would be desirable to provide details on which group the patients who received these treatments together were categorised.
OUR RESPONSE: We added the following sentence in the result section of the revised manuscript:
Lane 130-132: Patients who received treatment with combination of an anti-CD38 with PI/IMiD were categorized in the "anti-CD38 subgroup", because the anti-CD38 treatment was considered to have a particular impact on humoral responses post vaccination.
- Figure 5 was difficult to interpret and it was not clear what the number on the horizontal axis represented. The interpretation of the slope of the linear regression was also difficult to interpret.
OUR RESPONSE: Figure 5 shows correlations of neutralizing antibody to Wuhan, Delta, BA.1, BA2. and BA4/5, respectively, and binding antibody levels to Wuhan Spike-RBD (Panel A) and BA.2 (Panel B) of the 3 cohorts HCW (top panel), WM (middle panel) and MM (bottom panel).
We modified Figure 5 and we added the slope values of the linear regressions with Wuhan (red lettering) and BA4/5 (blue lettering). The slope p value is from the comparison of the linear regressions and which supports the significant differences among the two regressions. The steeper slopes of the BA.4/5 correlations showed the more rapid loss of NAb BA.4/5 compared to WA1.
Lane 403: …visualized by the steeper slopes.
- Furthermore, this report shows that most post-vaccination antibody titres, including HCW, have low neutralising activity against Omicrons. Considering that many infections with Omicorn variants have occurred in healthy individuals, who are also thought to have high antibody titres after booster, It is desirable to show the clinically relevant level of neutralizing titres protective against Omicron variants.
OUR RESPONSE: We previously showed that our assays binding Ab measured by ELISA and neutralizing Ab measured by the pseudotype virus assay directly correlated with live-virus neutralization and also correlated with protection in an macaque model (Rosati PLoS Pathog 17:e1009701; 2021). In this manuscript we show that many patients (60-80%) have very poor or lack of neutralizing Ab to Omicron variants, as summarized in Fig. 5F. Therefore, we concluded that
Lane 389-390:…Together, these data show that the patient cohorts are vulnerable with many patients not protected against Omicron.
Round 2
Reviewer 1 Report
The authors responded to my comments properly.